# Susceptibility-Weighted MRI for Predicting NF-2 Mutations and S100 Protein Expression in Meningiomas

**DOI:** 10.3390/diagnostics14070748

**Published:** 2024-03-31

**Authors:** Sena Azamat, Buse Buz-Yalug, Sukru Samet Dindar, Kubra Yilmaz Tan, Alpay Ozcan, Ozge Can, Ayca Ersen Danyeli, M. Necmettin Pamir, Alp Dincer, Koray Ozduman, Esin Ozturk-Isik

**Affiliations:** 1Institute of Biomedical Engineering, Bogazici University, Istanbul 34342, Turkey; 2Basaksehir Cam and Sakura City Hospital, Istanbul 34480, Turkey; 3Electrical and Electronics Engineering Department, Bogazici University, Istanbul 34342, Turkey; 4Department of Medical Biotechnology, Acibadem University, Istanbul 34752, Turkey; 5Department of Psychiatry and Neurochemistry, Institute of Neuroscience & Physiology, The Sahlgrenska Academy, University of Gothenburg, 42130 Mölndal, Sweden; 6Department of Biomedical Engineering, Acibadem University, Istanbul 34752, Turkey; 7Department of Medical Pathology, Acibadem University, Istanbul 34752, Turkey; 8Center for Neuroradiological Applications and Research, Acibadem University, Istanbul 34752, Turkey; 9Brain Tumor Research Group, Acibadem University, Istanbul 34752, Turkey; 10Department of Neurosurgery, Acibadem University, Istanbul 34752, Turkey; 11Department of Radiology, Acibadem University, Istanbul 34752, Turkey

**Keywords:** machine learning, meningioma, NF-2 mutations, S100 protein expression, susceptibility-weighted MRI

## Abstract

S100 protein expression levels and neurofibromatosis type 2 (NF-2) mutations result in different disease courses in meningiomas. This study aimed to investigate non-invasive biomarkers of NF-2 copy number loss and S100 protein expression in meningiomas using morphological, radiomics, and deep learning-based features of susceptibility-weighted MRI (SWI). This retrospective study included 99 patients with S100 protein expression data and 92 patients with NF-2 copy number loss information. Preoperative cranial MRI was conducted using a 3T clinical MR scanner. Tumor volumes were segmented on fluid-attenuated inversion recovery (FLAIR) and subsequent registration of FLAIR to high-resolution SWI was performed. First-order textural features of SWI were extracted and assessed using Pyradiomics. Morphological features, including the tumor growth pattern, peritumoral edema, sinus invasion, hyperostosis, bone destruction, and intratumoral calcification, were semi-quantitatively assessed. Mann–Whitney U tests were utilized to assess the differences in the SWI features of meningiomas with and without S100 protein expression or NF-2 copy number loss. A logistic regression analysis was used to examine the relationship between these features and the respective subgroups. Additionally, a convolutional neural network (CNN) was used to extract hierarchical features of SWI, which were subsequently employed in a light gradient boosting machine classifier to predict the NF-2 copy number loss and S100 protein expression. NF-2 copy number loss was associated with a higher risk of developing high-grade tumors. Additionally, elevated signal intensity and a decrease in entropy within the tumoral region on SWI were observed in meningiomas with S100 protein expression. On the other hand, NF-2 copy number loss was associated with lower SWI signal intensity, a growth pattern described as “en plaque”, and the presence of calcification within the tumor. The logistic regression model achieved an accuracy of 0.59 for predicting NF-2 copy number loss and an accuracy of 0.70 for identifying S100 protein expression. Deep learning features demonstrated a strong predictive capability for S100 protein expression (AUC = 0.85 ± 0.06) and had reasonable success in identifying NF-2 copy number loss (AUC = 0.74 ± 0.05). In conclusion, SWI showed promise in identifying NF-2 copy number loss and S100 protein expression by revealing neovascularization and microcalcification characteristics in meningiomas.

## 1. Introduction

Meningiomas are typically benign, slow-growing dural neoplasms that account for approximately 40% of all intracranial tumors [1]. Despite being recognized under one name, meningiomas are a heterogeneous group of tumors that differ in their anatomical locations, growth patterns, histopathologic findings, molecular genetic structures, epigenetic changes, responses to treatment, and outcomes [2]. Predicting meningioma behavior is crucial for treatment planning and prognosis, with recent studies primarily focusing on preoperatively determining the World Health Organization (WHO) grade, which correlates with an accurate prognosis [3,4,5]. However, molecular genetic studies have revealed that more than two-thirds of meningiomas carry deletions or mutations in the neurofibromatosis type 2 (NF-2) gene [6], which typically manifest as multiple tumors with a relatively poor prognosis, thus entailing an increased risk of mortality [7,8]. In this context, immunohistochemical studies have also been conducted to gain more insight into the anticipated behavior of meningiomas over time [9,10,11,12]. Some studies suggest that meningiomas with S100 protein expression tend to have a lower tumor grade and longer progression-free survival [10,12]. The purpose of these studies was to find cost-effective surrogate markers, like NF-2 mutations and S100 protein expression, that could help predict how meningiomas would behave. However, identifying these markers requires tissue diagnosis, which can be expensive and has limitations [6,13]. In response to this, a growing interest in the preoperative identification of surrogate markers in meningiomas for better treatment planning has recently been observed.

Neuroimaging is an essential tool for diagnosing brain tumors, planning treatments, assessing outcomes, and identifying early signs of tumor recurrence. In particular, magnetic resonance imaging (MRI) offers insights into tumor cell density, metabolic activity, and blood vessel formation. Susceptibility-weighted MRI (SWI) is an imaging technique that is sensitive to blood vessel formation and intratumoral calcification [14]. SWI utilizes magnetic susceptibility discrepancies resulting from differences in blood oxygen levels or calcification-dependent phase changes in venous blood, calcifications, or the surrounding brain parenchyma [15] to evaluate intratumoral susceptibility signals (ITSSs), which manifest as distinct linear or punctate areas with reduced signal intensity on images [16,17,18]. However, accurately quantifying ITSSs presents significant challenges due to the irregular characteristics of individual vessels and complex vascular networks. Nevertheless, this approach has shown promise in predicting glioma grades and molecular markers, such as isocitrate dehydrogenase (IDH) mutations and O-6-methylguanine-DNA methyltransferase methylation, before surgery [16,19,20]. Notably, neovascularization has been associated with a poor prognosis for brain tumors [21]. Additionally, NF-2-associated meningiomas often exhibit intratumoral calcification, which is also related to a poor prognosis [22]. This suggests that SWI, as the most sensitive imaging technique available for determining calcification and new blood vessel formation, might have the potential to predict meningioma behavior. This hypothesis is consistent with that of a study by Hu et al. [3], which demonstrated that integrating SWI into other MRI techniques significantly improved the accuracy of the noninvasive prediction of tumor grades in meningiomas.

Radiomics is an emerging field that enables the quantitative extraction of various features from medical images, including characteristics such as tumor shape, intensity, and texture, that serve as objective metrics for analysis [23]. In the case of a tumor, variations in radiomics features might reflect underlying heterogeneity [24]. Notably, radiomics has already exhibited the potential for distinguishing between low-grade and high-grade meningiomas [3,5,25,26]. However, traditional radiomics methods have found it difficult to capture the intrinsic characteristics of meningiomas due to considerable variability among their 15 histological subtypes, as established by the WHO classification. These distinct meningioma subtypes may display varying vascular, hemorrhagic, calcific, and fibrotic features, leading to significant heterogeneity regardless of their grade. In this context, deep learning-based feature extraction approaches, particularly those utilizing convolutional neural networks (CNNs), have gained prominence [4,27]. CNNs employ hierarchical convolution operations to extract valuable information about tumor characteristics, demonstrating enhanced predictive capabilities compared to traditional radiomic approaches. [27]. For instance, Zhu et al. [4] demonstrated that features derived from a deep learning architecture were more effective than traditional radiomics in assessing meningioma grading. Drawing on the above discussion, this study aimed to identify specific SWI markers to detect the presence of NF-2 mutations and S100 protein expression in meningiomas by utilizing both radiomics and deep-learning features.

## 2. Materials and Methods

### 2.1. Clinical Cohort and Data Acquisition

A total of 132 patients diagnosed with meningioma who had undergone preoperative MRI were included in this study. The study protocol was approved by the Bogazici University Science and Engineering Human Research Ethics Committee (03/2019). The inclusion criteria for the study cohort were as follows: (i) patients with an intracranial meningioma diagnosis, (ii) patients who underwent routine brain MRI examinations before surgical resection, and (iii) patients who underwent histopathological examination. The exclusion criteria were patients whose MRI images were of insufficient quality to enable proper assessment of the parameters (for example, owing to motion artifacts), and those who had more than one meningioma. Finally, 92 meningiomas with NF-2 mutation data and 99 with S100 protein expression data were selected for further analysis (Figure 1).

### 2.2. Histopathological and Genetic Analyses

A board-certified neuropathologist with 13 years of experience (A.E.D) reviewed all patient cases based on the current diagnostic criteria for assigning a World Health Organization (WHO) grade [28]. Subsequently, an NF-2 copy number variation test was performed to detect alterations in the number of copies of the NF-2 gene in the DNA isolated from the tumor or the blood sample of each patient. This study also assessed NF-2 allele loss in the meningioma tumor samples. The expected diploid status of the NF-2 copy number in a healthy individual was assumed to be 2, meaning that a sub-2 value indicated allelic loss, whereas a value above 2 suggested an increase in the copy number. Furthermore, a droplet digital polymerase chain reaction was conducted to detect NF-2 copy number variations (QX200 Droplet Digital PCR System, Bio-Rad Laboratories, Inc., Hercules, CA, USA). In addition, the tumor samples were subjected to immunohistochemical analysis using anti-S100 antibodies to obtain data on S100 protein expression.

### 2.3. MRI Protocol

A routine brain MRI protocol, which included pre- and post-contrast T1-weighted MRI (T1WI and ce-T1WI, respectively) (repetition time [TR]/echo time [TE] = 589/10 ms, field of view [FOV] = 220 mm, slice thickness = 3 mm), T2-weighted MRI (T2WI) (TR/TE = 3810/101 ms, FOV = 220 mm, slice thickness = 3 mm), fluid-attenuated inversion recovery (FLAIR) (TR/TE = 8320/92 ms, TI = 2000 ms, FOV = 220 mm, slice thickness = 3 mm), and SWI-MRI (TR/TE = 28/20 ms, FOV = 220 mm, slice thickness = 1.6 mm), was performed using a 3T clinical MRI scanner (Siemens Healthcare, Erlangen, Germany). After completing the pre-contrast series, a manual injection of 0.2 mL per kilogram of Gd-DTPA (Magnevist, Bayer HealthCare Pharmaceuticals, Wayne, NJ, USA) was administered to the patient, after which the post-contrast series was acquired. The dosage range for this injection varied from 7.5 mL to 15 mL depending on the patient’s weight. The MRI scanner software (Syngo XA30) automatically generated the magnitude and phase images, which were combined to obtain the post-processed SWI image. Subsequently, the phase, magnitude, and post-processed SWI images were recorded. A senior radiologist with 31 years of experience (A.D) reviewed all the MRI data, confirming that the images were acquired correctly, encompassed the tumor region, and exhibited a signal-to-noise ratio suitable for quantitative analysis.

### 2.4. Semi-Quantitative Evaluation of Morphological Characteristics

A neuroradiologist with 31 years of experience (A.D.) and a neurosurgeon with 18 years of experience (K.Ö.) evaluated all the MRI images without any prior access to the histopathological findings. They assessed the various morphological characteristics of the tumors, including the growth pattern, presence of peritumoral edema, dural venous sinus invasion, hyperostosis, bone destruction, and intratumoral calcification. A score of 1 was assigned to a “globose” growth pattern, indicating a rounded tumor growing inward from the dura, whereas a score of 0 was assigned to an “en plaque” growth pattern, indicating diffuse flat growth along the dura [29]. The growth pattern and dural venous sinus invasion were evaluated on ce-T1WI. Furthermore, the presence of peritumoral edema was assessed using FLAIR images. Hyperostosis and bone destruction were examined using T1WI. Although establishing a distinction between calcifications, primarily composed of calcium phosphate, and blood products in post-processed SWI images poses a challenge because both are displayed as signal drop-out and blooming effects, this was achieved in the current study using phase images, since calcification is diamagnetic and blood products are paramagnetic [30]. Figure 2 illustrates the identification of calcification and neovascularization on phase MR images.

### 2.5. Image Processing

All available anatomical MR series were employed for the purpose of delineating the tumor, peritumoral edema, cyst, and necrosis. Meningiomas are dural-based tumors that typically exhibit homogeneous contrast enhancement on ceT1-WI. Therefore, FLAIR images were used to delineate the complete tumor region, including the peritumoral edema and the solid tumor—observed as a bright region separate from the cerebrospinal fluid signal—using Slicer version 4.8.1 (http://slicer.org/ (accessed on 20 January 2024)). FLAIR images were selected specifically for their ability to highlight increased peritumoral edema, which serves as a surrogate marker for high-grade meningiomas [5,31,32,33]. Tumor delineation was conducted manually, slice-by-slice, using a semi-automated approach by two experienced individuals. Two readers independently performed lesion segmentations, and their results were cross-checked with each other for segmentation robustness. Following this, image registration was performed for each patient, which involved aligning the FLAIR images with high-resolution SWI using the FMRIB Software Library version 6.0.4 (FSL; http://fsl.fmrib.ox.ac.uk/fsl/fslwiki/ (accessed on 20 January 2024)). The transformation matrices derived from this registration were then used to align the segmentation masks for the tumor and the surrounding edema to the SWI for each patient. Following image registration, the post-processed SWI images and their corresponding segmentation masks were resampled to achieve a voxel size of 2 mm × 2 mm × 2 mm. Subsequently, the first-order textural features of the SWI within FLAIR hyperintensity were extracted using Pyradiomics v3.1.0—an open-source Python package [24].

### 2.6. Statistical Analysis

Mann–Whitney U tests were performed to compare the first-order textural features between the NF-2 copy number loss and the S100 protein expression groups. The *p* value for the Mann–Whitney U test was adjusted for multiple comparisons using the Bonferroni correction, resulting in an adjusted significance level of 0.0027 (0.05 divided by 18, representing the number of first-order textural features). Furthermore, both univariable and multivariable logistic regression analyses were performed to examine the associations between the first-order textural features, morphological features (*n* = 6), and age with regard to NF-2 mutations and S100 protein expression. The results of these analyses are reported as odds ratios (ORs), with corresponding 95% confidence intervals (CIs). The significance level of the *p* value for the univariable logistic regression analysis was set at 0.002 (0.05 divided by 24, which accounted for the sum of the number of first-order textural features, morphological features, and age) to address multiple comparisons, while that for the multivariable logistic regression was set at 0.05. To assess the multivariable logistic regression model’s performance and reduce overfitting, the data were split into training (80%) and test (20%) sets. Bootstrapping and cross-validation were applied to the training set, employing a stratified k-fold cross-validation strategy with five folds within the training set. ROC curves were generated for each bootstrap sample to illustrate the trade-off between true and false positive rates for different classification thresholds. This process was repeated iteratively to obtain multiple ROC curves. Furthermore, to estimate the variance of the ROC curves, the mean and standard deviation values of performance metrics such as the area under the ROC curve (AUC), accuracy, sensitivity, and specificity were calculated for each fold of the cross-validation procedure. Afterwards, the trained model was evaluated using the test set, which was held out during model training. Additionally, the influence of NF-2 copy number loss and S100 protein expression on categorical clinical variables, including WHO grade, intratumoral calcification, peritumoral edema, sinus invasion, bone destruction, hyperostosis, and the growth pattern, were assessed using 2 × 2 χ² tests and Fisher’s exact test when necessary. Lastly, principal component analysis (PCA) was employed to explore the relationship between demographic/radiological characteristics and molecular markers in understanding S100 protein expression variability and NF-2 copy number loss variability. The statistical analysis was conducted using MATLAB Statistics and Machine Learning Toolbox R2021a (MathWorks, Natick, MA, USA), the Statistical Package for the Social Sciences (SPSS version 27), and R (version 2022.12.0).

### 2.7. Deep Learning-Based Feature Extraction

Deep learning-based feature extraction involves utilizing the outputs from hidden layers of convolutional neural networks. To perform this task effectively, input images containing the tumor area were required. For each patient, the tumor area was enclosed within clearly defined rectangular boundaries on the images to meet the input requirements. Initially, a slice containing the largest tumoral area, along with its two consecutive slices, were selected. Subsequently, rectangular bounding boxes were delineated based on the x and y dimensions of the largest tumoral area on each slice. These regions were then cropped, scaled to an initial input size of 224 × 224 × 3, and preprocessed according to the model’s specifications [34]. For the NF-2 mutations dataset, 92 images were analyzed, and for the S100 protein expression dataset, 99 images were used as input. To mitigate overfitting, we adopted a transfer learning approach by leveraging a pretrained network for feature extraction [35]. Specifically, we utilized the pre-trained weights of ResNet50 to construct a custom classification algorithm [34]. Modifications were made to the ResNet50 model, which involved excluding the global average pooling, flattening, and dense connected layers, and including the global max pooling after each layer, with the aim of enhancing feature extraction capabilities while minimizing computational complexity and the risk of overfitting [34]. By excluding the global average pooling, flattening, and dense connected layers, the emphasis shifted towards directly extracting features from the convolutional layers of the modified ResNet50 model. The integration of global max pooling after each layer enabled the capture of the most significant features within each feature map, while retaining spatial information. Moreover, by aggregating the output of each layer, the modified ResNet50 model facilitated the extraction of hierarchical features at various levels. This hierarchical representation of features allowed the model to capture both low-level and high-level visual patterns inherent in the input images, thereby contributing to enhanced performance in subsequent tasks. These features were stored as tabular data. The modified deep learning network yielded 2048 features in its output.

Following the feature extraction process, the features were utilized as inputs for constructing machine learning models using the PyCaret framework [36]. First, the dataset was split into training and test sets. For the NF-2 related instances, 19 cases were designated as the test set. Similarly, for the S100 related instances, 20 cases were designated as the test set. Afterwards, stratified k-fold cross-validation with five folds was employed to ensure robust evaluation of the training set. Prior to splitting the training set, random shuffling was performed to prevent any ordering effects. A pipeline was constructed for each fold, comprising data preprocessing steps such as outlier removal, standardization, dimensionality reduction using principal component analysis, feature selection using SelectKBest, oversampling, and classifier training. This pipeline ensured consistent data transformations across folds.

In this study, the isolation forest technique was employed as part of our preprocessing procedure to identify and exclude outliers from the training set. We first constructed isolation forest models separately for NF-2-related instances and S100-related instances [37]. For each model, the contamination parameter was set to 0.05, corresponding to the proportion of outliers aimed to be excluded. This led to the elimination of four NF-2-related instances from the training set, while an additional four instances related to S100 protein expression were also discarded. Afterwards, the features were standardized into structured data with zero mean and unit variance [38]. The scaler was fitted to the training set, which computed the mean and standard deviation of each feature and standardized the data. Subsequently, the same scaler instance was used to transform the test set.

A two-step feature selection process was initiated. First, PCA was applied to reduce the dimensionality of the features [39]. For the NF-2 copy number loss dataset, 2048 features were reduced to 73. For the S100 protein expression dataset, 2048 features were reduced to 91. Then, the select-from-model technique, based on the extreme gradient boosting algorithm in scikit-learn, was employed to select features for further analysis to optimize the classifier performance [40]. Following exploratory data analysis (EDA), we made informed decisions to limit the number of features. Specifically, the feature set was restricted to 18 for the S100 protein expression dataset and 11 for the NF-2 copy number loss dataset for each fold. This decision was guided by a thorough examination of the dataset, including correlation analysis, to ensure that only the most relevant and informative features were retained for modeling. By striking a balance between model complexity and performance, we aimed to enhance the interpretability and generalizability of our models.

After preprocessing and feature selection, the training set was oversampled using the Synthetic Minority Oversampling Technique (SMOTE) to address class imbalance [41]. SMOTE is a widely used method for handling imbalanced datasets by generating synthetic samples of the minority class to rebalance the class distribution. The SMOTE algorithm works by creating synthetic samples along the line segments connecting similar instances of the minority class. This is achieved by randomly selecting a minority class instance and its k nearest neighbors, and then generating new samples along the line segments joining these instances. For the NF-2 copy number loss dataset, after the removal of outliers and application of SMOTE, the training set originally comprising 69 cases was increased to 72 cases. Among the original 69 cases, 33 cases were classified as NF-2 intact, while 36 cases exhibited NF-2 mutations. The application of SMOTE led to a more balanced dataset, contributing to the total number of cases in the training set reaching 72. On the other hand, for the S100 protein expression dataset, the training set initially contained 75 cases. After applying SMOTE, the number of cases increased to 90.

In this study, we evaluated 12 different machine learning methods, encompassing both traditional and ensemble approaches. The traditional methods included linear and quadratic discriminant analysis [42], logistic regression [43], decision tree classifier [44], and naive Bayes [45]. Among the ensemble methods, we employed the AdaBoost classifier [46], also known as the adaptive boosting classifier, and the gradient boosting classifier [47], which is a generalization of AdaBoost. Additionally, we utilized the light gradient boosting machine (LGBM) [48], a tree-based learning algorithm within the gradient boosting framework. We also employed extreme gradient boosting [49], or XGBoost, due to its superior performance and scalability. Furthermore, we utilized the random forest classifier [50], a basic ensemble learning approach. Lastly, we employed the extra trees classifier [51], which is akin to random forest.

To further optimize the performance of our machine learning models, we utilized the tune_model function from the PyCaret framework. This function automatically tunes hyperparameters using a grid search approach within the specified search space, aiming to maximize the F1-score within each fold. Following hyperparameter optimization, performance metrics, including the F1-score and confusion matrices, were computed for each fold and averaged to assess overall model performance. The mean and standard deviation of these metrics were calculated to provide a comprehensive evaluation of the model’s performance.

Subsequently, the best machine learning algorithm was selected based on accuracy, area under the curve (AUC), sensitivity, and specificity metrics on the test set. Figure 3 illustrates the proposed deep learning-based feature extraction pipeline. The software codes used in this study can be found at https://github.com/Computational-Imaging-LAB/meningiomaSWI (accessed on 20 January 2024).

## 3. Results

### 3.1. Demographic and Radiological Characteristics

In a cohort of 92 patients with available NF-2 mutations data, 48 exhibited a loss of at least one NF-2 allele. Meanwhile, among the 99 patients with available S100 protein expression data, 39 displayed S100 protein expression. The mean age of the cohort was 52.9 ± 13.9 for patients with meningiomas with NF-2 mutations data and 52.3 ± 13.8 for those with S100 protein expression data. Notably, the age distribution did not differ significantly between the two subgroups for NF-2 mutations (*p* = 0.17) or those for S100 protein expression (*p* = 0.33) (Figure 4). The results further revealed that among the 39 meningiomas expressing the S100 protein (S100 (+)), 22 were high-grade tumors (WHO grade 2 + 3). In contrast, among the 60 meningiomas without any S100 protein expression (S100 (−)), 34 were high-grade tumors. Effectively, this analysis failed to reveal any significant differences in the WHO grade between the S100 protein expression subgroups (*p* = 0.96). However, 33 of the 48 meningiomas with NF-2 copy number loss were high-grade tumors, while only 19 of the 44 meningiomas with NF-2 wild type were high-grade tumors (*p* = 0.01). This suggests that NF-2 copy number loss is associated with an increased risk of a higher tumor grade. Notably, the χ² test and Fisher’s exact test could not identify any significant associations between NF-2 mutations or S100 protein expression and various morphological features, including peritumoral edema (*p* = 0.23 for NF-2 mutations, *p* = 0.20 for S100 protein expression), bone destruction (*p* = 0.99 for NF-2 mutations, *p* = 0.47 for S100 protein expression), sinus invasion (*p* = 0.62 for NF-2 mutations, *p* = 0.79 for S100 protein expression), hyperostosis (*p* = 0.81 for NF-2 mutations, *p* = 0.19 for S100 protein expression), the growth pattern (*p* = 0.90 for NF-2 mutations, *p* = 0.82 for S100 protein expression), and intratumoral calcification (*p* = 0.30 for S100 protein expression). Interestingly, a significant relationship between intratumoral calcification and NF-2 copy number loss was observed (*p* = 0.037) (Table 1, Figure 5).

Regarding SWI signal intensity values within the FLAIR hyperintensity, the S100 (+) group exhibited significantly higher values (−78.36, range = [−130.21–36.67]) compared to the S100 (−) group (−97.44, range = [−143.72–2.85]) (*p* < 0.001) (Appendix A). However, there were no significant differences in first-order textural features between the NF-2 mutations subgroups (Appendix A). Moreover, the univariable analysis failed to identify any variables that significantly predicted NF-2 copy number loss (Appendix A). In contrast, the multivariable analysis revealed that decreased maximum signal intensity of the tumor, “en plaque growth”, and the presence of intratumoral calcification were significant predictors of NF-2 copy number loss, with odds ratios of 0.98 (*p* = 0.015), 0.20 (*p* = 0.023), and 5.39 (*p* = 0.021), respectively (Appendix A). Furthermore, in the context of S100 protein expression, a higher minimum signal intensity value of SWI within FLAIR hyperintensity emerged as a predictor, with an odds ratio of 1.03 (*p* = 0.002), as indicated by the results of the univariable logistic regression analysis (Appendix A). Notably, entropy emerged as a significant predictor of S100 protein expression in the multivariable analysis, with an odds ratio of 2.16 × 10^−6^ (*p* = 0.049) (Appendix A). When evaluated as machine learning algorithms with cross-validation, the multivariable logistic regression model demonstrated a mean AUC of 0.79 ± 0.03 for predicting NF-2 copy number loss (accuracy = 0.49 ± 0.003, sensitivity = 0.48 ± 0.11, specificity = 0.48 ± 0.07) and a mean AUC of 0.83 ± 0.03 for identifying S100 protein expression (accuracy = 0.62 ± 0.002, sensitivity = 0.58 ± 0.17, specificity = 0.59 ± 0.07) on the training set. Additionally, the model achieved an accuracy of 0.59 for predicting NF-2 copy number loss (sensitivity = 0.57, specificity = 0.58) and an accuracy of 0.70 for identifying S100 protein expression (sensitivity = 0.38, specificity = 0.92) on the test set (Figure 6). PCA analysis revealed that specific features played a crucial role in explaining the variability observed in these molecular markers. For S100 protein expression variability, entropy emerged as significant contributors. In contrast, for NF-2 copy number loss variability, a broader set of features demonstrated significance. Notably, intratumoral calcification and image intensity distribution metrics such as the 90th percentile were identified as significant contributors for NF-2 copy number loss variability (Appendix A). Figure 7 and Figure 8 illustrate sample meningioma cases for NF2-L and NF2-NL, as well as the S100+ and S100− groups (Appendix A).

### 3.2. Diagnostic Performance of Deep Learning Models

Based on the evaluation of various algorithms for NF-2 copy number loss prediction, the LGBM model exhibited the best overall performance on the training set with cross-validation. Conversely, when considering S100 protein expression, the LGBM model demonstrated the best performance across most parameters, except for precision and AUC, where the extra trees classifier achieved superior results on the training set with cross-validation. For predicting S100 protein expression, the LGBM model utilized a feature set of 18 (Appendix A), achieving an accuracy of 0.78 ± 0.08 and an AUC of 0.78 ± 0.002. On the other hand, the extra trees classifier achieved an accuracy of 0.72 ± 0.11 and an AUC of 0.80 ± 0.12. The performance metrics for LGBM included a recall (sensitivity) of 0.77 ± 0.09, precision of 0.70 ± 0.09, and an F1-score of 0.74 ± 0.08. For the extra trees classifier, the corresponding metrics were a recall of 0.46 ± 0.22, precision of 0.79 ± 0.19, and an F1-score of 0.54 ± 0.19. In predicting NF-2 copy number loss, LGBM utilized a feature set of 11 (Appendix A) and achieved an accuracy of 0.75 ± 0.13, with an AUC of 0.84 ± 0.10. The performance metrics for LGBM included a recall of 0.70 ± 0.17, precision of 0.79 ± 0.11, and an F1-score of 0.74 ± 0.13 (Table 2 and Table 3).

Among the various machine learning algorithms used to predict S100 protein expression and NF-2 copy number loss, LGBM emerged as the most successful classifier for both tasks on the test set. To predict S100 protein expression, LGBM employed a set of 18 features, achieving a high accuracy of 0.80 and an AUC of 0.85 (sensitivity = 0.87, specificity = 0.75). In predicting NF-2 copy number loss, LGBM utilized a set of 11 selected features, resulting in an accuracy of 0.73 and an AUC of 0.74 (sensitivity = 0.70, specificity = 0.77). Without using SMOTE in the preprocessing stage, the performance of LGBM decreased. For predicting NF-2 copy number loss, LGBM resulted in an accuracy of 0.63 and an AUC of 0.67 (sensitivity = 0.70, specificity = 0.55). For predicting S100 protein expression, LGBM resulted in an accuracy of 0.75 and an AUC of 0.76 (sensitivity = 0.62, specificity = 0.83) (Appendix A). Lists of the features considered in predicting NF-2 copy number loss and S100 protein expression statuses are provided in Appendix A. Figure 9 shows the learning curves for the top-performing machine learning algorithm when using SMOTE. As depicted, the curves for each group plateau during testing. In contrast, Appendix A presents the learning curves for the same algorithm without SMOTE.

## 4. Discussion

This study demonstrated the potential of features extracted from SWI in predicting S100 protein expression and NF-2 mutations in meningiomas. The results demonstrated that changes in signal intensity values within the tumor area in SWI are significant indicators for both S100 protein expression and NF-2 mutations. Furthermore, specific tumor characteristics, such as intratumoral calcification and the “en plaque” growth pattern, emerged as vital predictors of NF-2 copy number loss.

Furthermore, the analysis of WHO grades in relation to S100 protein expression and NF-2 mutations offered interesting findings. While there was no significant difference in terms of WHO grade between the S100 protein expression groups, NF-2 copy number loss was observed to be associated with a higher risk of a high tumor grade. This implies that NF-2 copy number loss may serve as an indicator of more aggressive meningiomas, consistent with previous research [22]. Moreover, these findings highlight the importance of accounting for genetic factors in tumor grading and prognosis [10].

SWI has already demonstrated high sensitivity to neovascularization, bleeding, and intratumoral calcification in comparison to conventional MRI sequences, which it achieves by leveraging magnetic susceptibility discrepancies arising from phase differences within venous blood, calcifications, and surrounding brain tissue [15]. Additionally, previous studies have explored the potential of SWI for predicting meningioma grades [3,4,5]. For instance, a study by Hu et al. [3], which investigated the use of the radiomic features of a multiparametric MRI, including SWI, for preoperative tumor grade prediction, proposed that incorporating SWI radiomic features with apparent diffusion coefficient mapping and ce-T1W images could improve diagnostic performance [3]. An accuracy of 0.76 was achieved by using only SWI radiomics features to differentiate between high-grade and low-grade meningiomas, while the radiomics features derived from multiparametric MRI achieved an accuracy of 0.82. These results were consistent with the findings of the current study, in which the improved accuracy achieved was attributed to SWI’s ability to capture tumor heterogeneity arising from neovascularization [3].

Additionally, Zhang et al. [5] used morphological features, semi-quantitative SWI, and quantitative susceptibility mapping (QSM) features to distinguish between high- and low-grade meningiomas, finding that semi-quantitative SWI and quantitative QSM features did not significantly improve differentiation. However, morphological features, such as peritumoral edema, irregular tumor borders, and tumor location along cerebral convexity, proved to be valuable in predicting high-grade meningiomas. In the current study, although peritumoral edema and tumor location were not found to be predictors of differentiation, the growth pattern associated with the tumor border was identified as a predictor, consistent with the findings of Zhang et al. [5]. Previous studies indicated peritumoral edema as a surrogate marker for high-grade meningiomas [5,31,32,33]. This information led to the use of FLAIR images for the segmentation task in order to avoid missing any extra information within the peritumoral edema [5,31,32,33].

S100 proteins constitute a family of calcium-binding proteins whose biological functions include cell proliferation, apoptosis, motility, and cytoskeletal organization in specific cell types, primarily nerve, glial, and epithelial cells [52]. Notably, changes in S100 protein expression levels have a significant impact on tumor prognosis [10,12]. This study found that S100 protein expression does not exhibit a clear positive or negative association with the WHO grade. In contrast, a study conducted by Nassiri et al. involving 121 patients with meningiomas found that S100 protein expression was predominantly observed in low-grade tumors [10]. Additionally, in cases of completely resected WHO grade 1 meningioma, the risk of recurrence was found to be negatively correlated with the level of S100 protein expression [12]. Based on these earlier findings, it was hypothesized that S100 protein expression might be linked to the favorable prognostic imaging features of SWI, even if it did not directly correlate with the WHO grades of our limited patient cohort. Notably, by investigating this issue, this study became the first to explore the correlation between S100 protein expression and imaging features using SWI. The findings of this study revealed that S100 protein expression was associated with decreased entropy and increased signal intensity values within the tumor on SWI. Higher entropy in imaging typically indicates greater heterogeneity within a tumor. In the context of SWI, this heterogeneity may be attributed to various factors, including neovascularization and microcalcification, which produce linear or dot-like areas of low signal scattering [16]. These results suggest that meningiomas with S100 protein expression may exhibit greater homogeneity, appear brighter on imaging, and have fewer features, such as neovascularization, calcification, and bleeding, compared to those without such expression. Although S100 protein expression did not directly correlate with the WHO grade, the observed tumor homogeneity and brightness suggest decreased aggressiveness, which was consistent with the findings of Nassiri et al. [10]. The machine learning results, based on the pretrained models, further confirmed that S100 protein expression influenced the appearance of tumors on SWI images. Notably, the model was able to distinguish S100 protein expression in meningiomas with relatively high accuracy.

NF-2 is a tumor suppressor gene located on chromosome 22q12 that encodes the Merlin protein [53], which plays a crucial role in stabilizing the cell membrane’s cytoskeleton [54]. Several studies conducted on patients with NF-2, including those suffering from skin cancers, vestibular schwannomas, and meningiomas, have posited that the loss of heterozygosity, including allelic loss of NF-2, may contribute to neoplastic developments [55,56]. This study identified NF-2 copy number loss as a significant risk factor for a high WHO grade. Notably, this finding was consistent with prior research indicating that individuals with NF-2 mutations tend to develop meningiomas characterized by higher grades, worse prognoses, and increased recurrence rates compared to those without these mutations [57,58]. Based on the scope of this study, it was discerned that the occurrence of NF-2 copy number loss can be reasonably anticipated by conducting a nuanced evaluation of certain parameters. Specifically, the signal intensity values of SWI, the distinctive “en plaque” growth pattern exhibited by tumors, and the presence of intratumoral calcification emerged as significant indicators bearing predictive value. Consistent with these findings, prior research has also observed that meningiomas with NF-2 mutations exhibit some distinctive traits, including the presence of psammoma bodies and a significantly elevated mitotic index compared to sporadic meningiomas [22]. Psammoma bodies, which appear as concentric, lamellated, and calcified structures within a tumor, are among the most salient features observed in NF-2 mutation-associated meningiomas. This correlation between NF-2 mutations and psammoma bodies underscores the potential diagnostic and prognostic significance of such histological attributes in distinguishing between meningiomas based on their imaging features on SWI [59]. Furthermore, the presence of intratumoral calcification helped elucidate the relationship between decreased signal intensity values in SWI-MRI and NF-2 copy number loss. This study also identified that the “en plaque” growth pattern, typically seen in tumors with a high tumor grade, is associated with NF-2 copy number loss. This finding was consistent with the studies conducted by Morin et al. [60] and Zhu et al. [4], who investigated 314 meningiomas and 181 meningiomas, respectively, and recognized a similar association between low sphericity and high tumor grade. Overall, the pretrained base machine learning model predicted NF-2 mutations in meningiomas using SWI images with an accuracy that was lower than that for the prediction of S100 protein expression in this limited patient cohort.

The use of deep learning in brain tumor image analysis, including tumor localization, segmentation, and identification of tumor subtypes, has been well documented [61,62]. This particular study aimed to explore the potential of deep learning networks, comprising numerous self-learning units, to quantify prognostic features associated with the molecular characteristics of meningiomas. Additionally, this study investigated the use of radiomics and morphological features to predict the molecular characteristics of meningiomas. Our results indicated that the radiomics and morphological features were useful in identifying molecular subgroups of meningiomas. This finding aligned with those of Zhang et al. [5], who successfully identified high-grade meningiomas by considering morphological features, such as peritumoral edema, irregular tumor borders, and tumor location along the cerebral convexity, alongside SWI imaging features. Furthermore, our findings were consistent with a previous study, indicating that the deep learning-based feature extraction effectively captured the heterogeneity of meningiomas and distinguished between their molecular types [4]. In their work, deep learning features were extracted using the Xception network, followed by dimensionality reduction using the random forest algorithm and sequential backward selection method. Subsequently, linear discriminant analysis was employed for classification purposes. In our study, we employed a modified version of the ResNet50 architecture. By implementing global max pooling after each layer, we aimed to capture the most significant features within each feature map while retaining spatial information, which is crucial for accurate image classification tasks. This modification allowed our model to extract hierarchical features at multiple levels, thereby capturing both low-level and high-level visual patterns present in the input images. Additionally, we adopted 14 different classifier methods to explore various avenues for improving classification performance. While direct comparisons with state-of-the-art (SOTA) models were not feasible in our study due to differences in methodologies and datasets, our approach built upon existing techniques and addressed specific challenges in image classification. Our results demonstrated the efficacy of modified ResNet50 architecture and the diverse ensemble of classifier methods employed in achieving a competitive performance for the tasks at hand.

This study has certain limitations. First, the sample size was relatively small, comprising 92 patients with the NF-2 mutation dataset and 99 with the S100 protein expression dataset. Therefore, more extensive and more expansive studies must be conducted to validate and strengthen the findings of this study. Additionally, the MRI images considered in this study were obtained from a single institution, and external validation using data from multiple institutions with varying imaging protocols was not performed. This could limit the generalizability of the study results. Another limitation of this study was the exclusion of MRI images generated by protocols other than SWI from the analysis, which resulted in a dataset solely focused on SWI. While SWI is valuable for assessing neovascularization and microcalcifications [14], it is essential to recognize that calcification and neovascularization exhibit different characteristics on phase images [30]. In meningiomas, tumors with NF-2 mutations tend to develop calcifications, whereas those with S100 protein expression may not show neovascularization or calcification [10,22]. The decision to restrict the analysis to SWI was made to prioritize consistency and concentrate specifically on its capabilities in identifying microcalcifications and neovascularization. Alternative MRI protocols may lack the same level of sensitivity or specificity for these features. We aimed to maintain homogeneity within the dataset and enable a more precise evaluation of SWI’s effectiveness in detecting these critical features in meningiomas. This focused approach ensured a more accurate assessment of SWI’s performance in this context. Moreover, although the deep learning model utilized in this study exhibited promising diagnostic potential, further evaluation and refinement are necessary to improve its accuracy. Although the modified ResNet50 model, when used alongside an LGBM classifier, demonstrated a strong performance in predicting S100 protein expression, its accuracy in predicting NF-2 copy number loss was relatively lower. On the other hand, considering that the CNN used in this study was based on conventional architecture, employing more advanced and novel network architectures, such as graph convolutional networks [63] or complementary attention algorithms [64], in addition to CNN, could yield more specific results. Additionally, the proposed model did not account for some significant parameters, such as clinical characteristics, that could influence the non-invasive diagnosis of NF-2 mutations. Finally, the imbalanced distribution of classes in the dataset posed a challenge for machine learning algorithms when conducting image classification. Although SMOTE was implemented to balance the classes, it is essential to recognize that this could have impacted the results.

In conclusion, this study demonstrated the potential of SWI-based imaging features in predicting S100 protein expression and NF-2 mutations in meningiomas. Changes in signal intensity values within the tumor area in the SWI emerged as significant indicators of both S100 protein expression and NF-2 mutations. Furthermore, specific tumor characteristics, such as intratumoral calcification and the “en plaque” growth pattern, were identified as vital factors for predicting NF-2 copy number loss. While deep learning models, radiomics, and morphological features all showed promise in predicting these molecular characteristics, their performances were comparable. Moreover, this study highlighted the potential of SWI as a non-invasive tool for predicting molecular characteristics in meningiomas, thus offering valuable insights for treatment planning. Further research and validation are necessary to maximize the utilization of SWI’s potential in clinical settings.

## Figures and Tables

**Figure 1 diagnostics-14-00748-f001:**
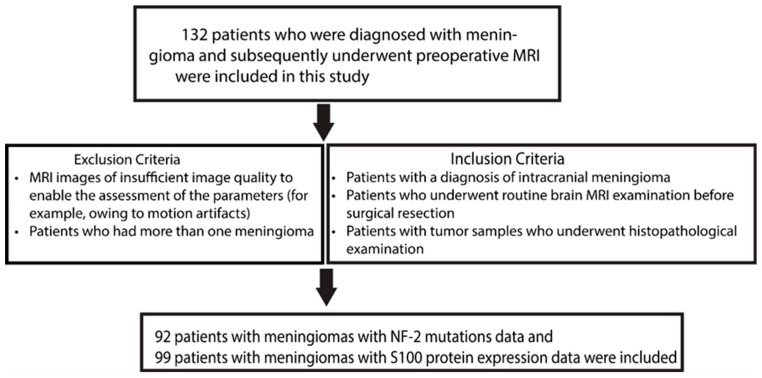
The flow diagram of the patient selection process.

**Figure 2 diagnostics-14-00748-f002:**
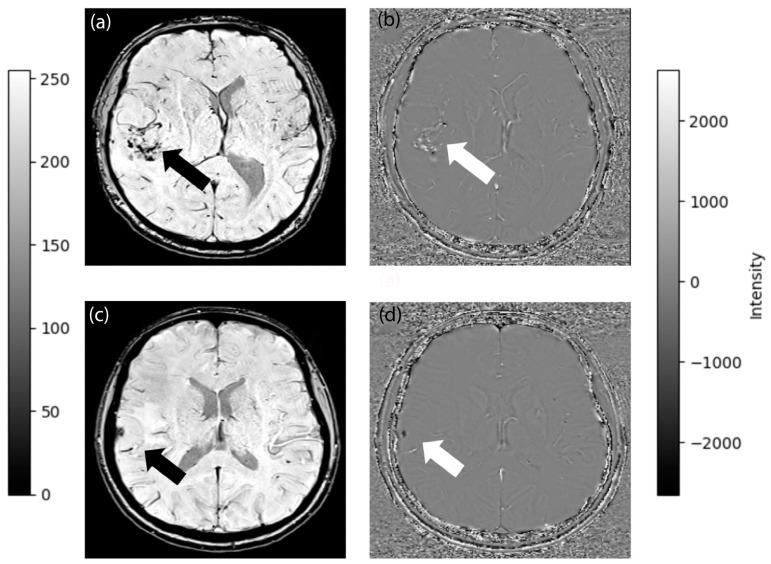
SWI-MRI (**a**) and phase image (**b**) of a meningioma with neovascularization (dark in SWI and white in phase image). SWI-MRI (**c**) and phase image (**d**) of a meningioma that has calcification (dark in both SWI and phase image). Black arrows depict signal void regions on SWI-MRI. White arrows depict phase difference regions within the tumor.

**Figure 3 diagnostics-14-00748-f003:**
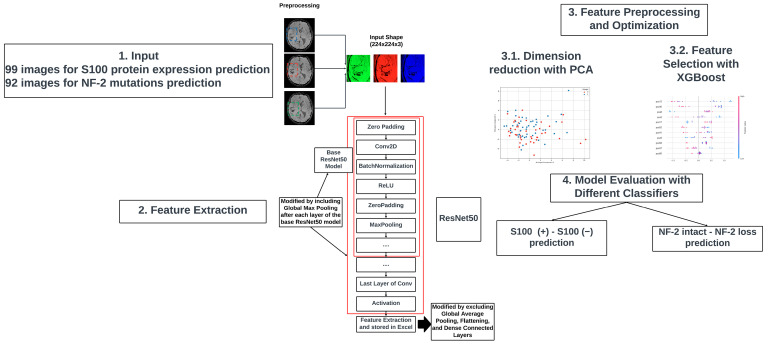
The pipeline of the proposed deep learning-based feature extraction.

**Figure 4 diagnostics-14-00748-f004:**
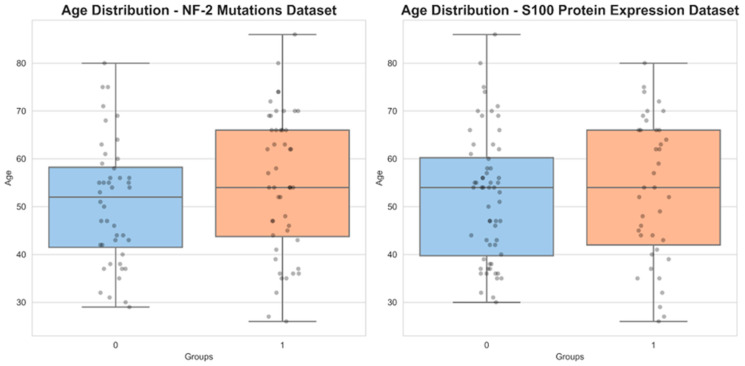
Age distribution of NF-2 mutations and S100 protein expression datasets. The data distributions are represented as median and interquartile ranges (IQRs).

**Figure 5 diagnostics-14-00748-f005:**
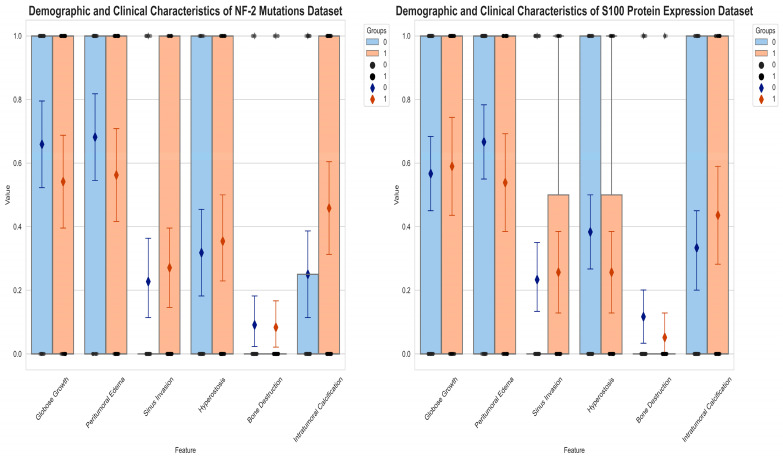
Demographic and clinical characteristics of NF-2 mutations and S100 protein expression datasets. The data distributions are represented as median and interquartile ranges (IQRs).

**Figure 6 diagnostics-14-00748-f006:**
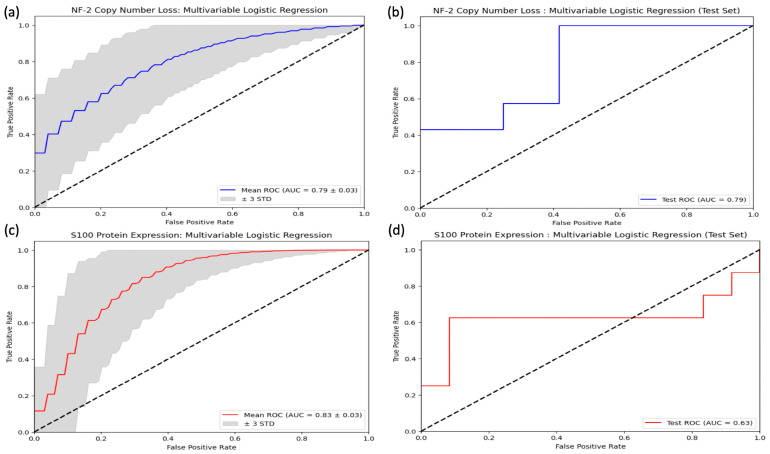
Receiver Operating Characteristic (ROC) curves for the performance of the multivariable logistic regression models in distinguishing between cases with and without NF-2 copy number loss and S100 protein expression. (**a**) Cross-validation results within training set of NF-2 copy number loss, (**b**) the test set result of NF-2 copy number loss, (**c**) cross-validation results within training set of S100 protein expression, (**d**) the test set result of S100 prote.in expression.

**Figure 7 diagnostics-14-00748-f007:**
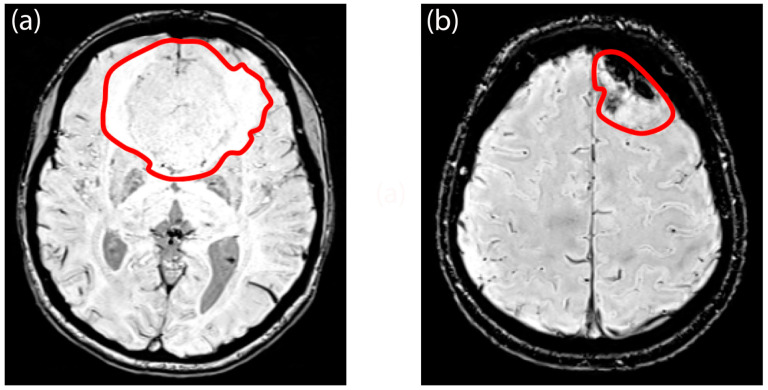
Example tumor region of interest of (**a**) a meningioma with NF-2 wild type and (**b**) a meningioma with NF-2 copy number loss. The patient with NF-2 copy number loss demonstrates intratumoral calcification and an “en plaque” growth pattern.

**Figure 8 diagnostics-14-00748-f008:**
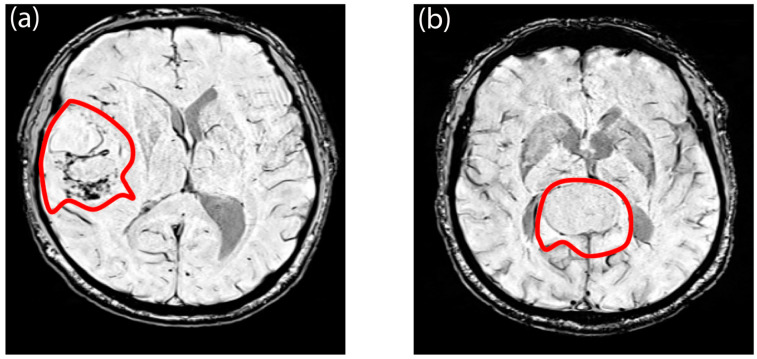
Example tumor region of interest of (**a**) a meningioma which did not show S100 protein expression and (**b**) a meningioma with S100 protein expression. The meningioma without the S100 protein expression depicts more tumor heterogeneity.

**Figure 9 diagnostics-14-00748-f009:**
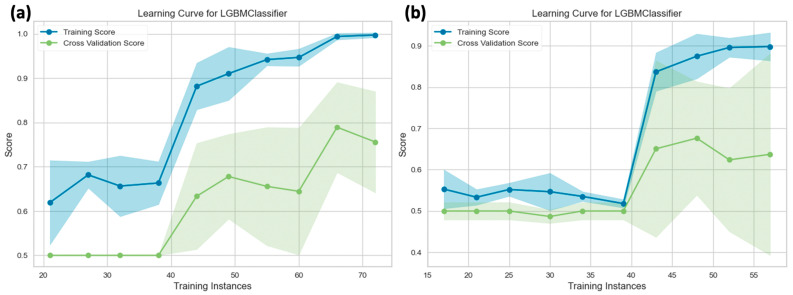
The learning curves and cross-validation scores on the training dataset for the (**a**) S100 protein expression group (number of training instances = 90, fold number = 5), and (**b**) NF-2 mutation group (number of training instances = 72, fold number = 5). The uncertainty bands in the learning curves depict the standard deviations of the cross-validated score.

**Table 1 diagnostics-14-00748-t001:** Morphological features and WHO grades of meningiomas.

Molecular Analysis	High Grade ^a^	Peritumoral Edema ^a^	Bone Destruction ^b^	Hyperostosis ^a^	Sinus Invasion ^a^	Tumoral Calcification ^a^	Globose Growth ^a^
NF-2 Loss	33/48	26/48	4/48	17/48	13/48	22/48	26/48
NF-2 Intact	19/44	13/44	4/44	14/44	10/44	11/44	29/44
S100 (+)	22/39	21/39	2/39	10/39	10/39	17/39	23/39
S100 (−)	34/60	40/60	7/60	23/60	14/60	20/60	34/60

High-grade: WHO grades 2 and 3 based on the WHO 2021 grading system for meningiomas. Peritumoral Edema: Refers to the swelling of the surrounding brain tissue caused by the tumor’s presence. Bone Destruction: Indicates the erosion or destruction of adjacent bone structures by the tumor. Hyperostosis: Suggests excessive bone growth or thickening in response to the tumor. Sinus Invasion: Signifies the tumor’s involvement with the venous sinuses in the brain. Intratumoral Calcification: Refers to the presence of calcified deposits within the tumor. Globose Growth: Describes a rounded, ball-like growth pattern extending inward from the dura mater. En Plaque Growth: Indicates a diffuse, flat, or sheet-like growth pattern along the dura mater. ^a^ Comparison of morphological features and WHO grades using χ^2^ tests. ^b^ Comparison of morphological features using Ficher’s exact test.

**Table 2 diagnostics-14-00748-t002:** Performance comparison of machine learning algorithms for NF-2 copy number loss prediction. ** indicates the best performance on the training set.

Algorithms	AUC	Accuracy	Recall	Precision	F1	Kappa
Light Gradient Boosting Machine **	0.84 ± 0.10	0.75 ± 0.13	0.70 ± 0.17	0.79 ± 0.11	0.74 ± 0.13	0.51 ± 0.26
Extreme Gradient Boosting	0.78 ± 0.05	0.68 ± 0.09	0.63 ± 0.05	0.75 ± 0.16	0.68 ± 0.07	0.37 ± 0.19
Extra Trees Classifier	0.65 ± 0.11	0.57 ± 0.11	0.67 ± 0.16	0.58 ± 0.08	0.61 ± 0.10	0.15 ± 0.23
Logistic Regression	0.65 ± 0.10	0.61 ± 0.10	0.66 ± 0.11	0.66 ± 0.18	0.64 ± 0.06	0.21 ± 0.21
Random Forest Classifier	0.62 ± 0.12	0.56 ± 0.07	0.64 ± 0.23	0.57 ± 0.05	0.58 ± 0.11	0.12 ± 0.16
AdaBoost Classifier	0.61 ± 0.19	0.56 ± 0.18	0.65 ± 0.32	0.53 ± 0.16	0.57 ± 023	0.14 ± 0.35
Gradient Boosting Classifier	0.60 ± 0.07	0.59 ± 0.10	0.58 ± 0.18	0.66 ± 0.19	0.58 ± 0.11	0.19 ± 0.21
Linear Discriminant Analysis	0.58 ± 0.15	0.53 ± 0.03	0.60 ± 0.10	0.54 ± 0.04	0.56 ± 0.04	0.06 ± 0.09
K-Neighbors Classifier	0.54 ± 0.20	0.53 ± 0.12	0.66 ± 0.12	0.53 ± 0.11	0.59 ± 0.11	0.05 ± 0.24
Naive Bayes	0.53 ± 0.13	0.55 ± 0.15	0.54 ± 0.15	0.61 ± 0.21	0.55 ± 0.12	0.09 ± 0.30
Decision Tree Classifier	0.53 ± 0.08	0.52 ± 0.08	0.60 ± 0.11	0.54 ± 0.08	0.56 ± 0.07	0.05 ± 0.16
Quadratic Discriminant Analysis	0.49 ± 0.09	0.49 ± 0.09	0.50 ± 0.20	0.48 ± 0.14	0.49 ± 0.17	−0.01 ± 0.20

**Table 3 diagnostics-14-00748-t003:** Performance comparison of machine learning algorithms for S100 protein expression. ** indicates the best performance on the training set.

Algorithms	AUC	Accuracy	Recall	Precision	F1	Kappa
Extra Trees Classifier	0.80 ± 0.12	0.72 ± 0.11	0.46 ± 0.22	0.79 ± 0.19	0.54 ± 0.19	0.38 ± 0.24
Naive Bayes	0.79 ± 0.11	0.69 ± 0.06	0.37 ± 0.17	0.77 ± 0.20	0.47 ± 0.15	0.29 ± 0.15
Linear Discriminant Analysis	0.78 ± 0.15	0.71 ± 0.12	0.61 ± 0.28	0.70 ± 0.19	0.59 ± 0.21	0.39 ± 0.27
Light Gradient Boosting Machine **	0.78 ± 0.002	0.78 ± 0.08	0.77 ± 0.09	0.70 ± 0.09	0.74 ± 0.08	0.55 ± 0.15
Logistic Regression	0.75 ± 0.13	0.68 ± 0.11	0.56 ± 0.24	0.58 ± 0.13	0.55 ± 0.20	0.32 ± 0.25
K-Neighbors Classifier	0.72 ± 0.12	0.45 ± 0.04	0.96 ± 0.06	0.41 ± 0.02	0.57 ± 0.03	0.06 ± 0.08
Random Forest Classifier	0.72 ± 0.11	0.63 ± 0.06	0.38 ± 0.20	0.45 ± 0.23	0.40 ± 0.21	0.19 ± 0.13
Extreme Gradient Boosting	0.64 ± 0.16	0.64 ± 0.10	0.48 ± 0.10	0.58 ± 0.21	0.51 ± 0.12	0.24 ± 0.21
Quadratic Discriminant Analysis	0.56 ± 0.13	0.58 ± 0.04	0.10 ± 0.08	0.35 ± 0.37	0.15 ± 0.12	−0.01 ± 0.12
AdaBoost Classifier	0.54 ± 0.10	0.53 ± 0.10	0.37 ± 0.16	0.41 ± 0.15	0.37 ± 0.15	0.02 ± 0.20
Decision Tree Classifier	0.51 ± 0.10	0.52 ± 0.11	0.52 ± 0.07	0.41 ± 0.07	0.46 ± 0.07	0.03 ± 0.19
Gradient Boosting Classifier	0.45 ± 0.05	0.52 ± 0.06	0.52 ± 0.18	0.41 ± 0.05	0.44 ± 0.07	0.03 ± 0.11

## Data Availability

Data is unavailable due to ethical restrictions.

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
