# Peer review of "Susceptibility-Weighted MRI for Predicting NF-2 Mutations and S100 Protein Expression in Meningiomas"

_diagnostics, 2024, doi:10.3390/diagnostics14070748_

Round 1

Reviewer 1 Report

Comments and Suggestions for Authors

The manuscript by Azamat et al. describes the adaptation of the classic DL-based tool ResNet50 for extraction of deep learning features characteristics, which was applied for SWI-modality MRIs shots of meningiomas to discriminate between radiological characteristics and molecular markers.

The modified image processing ResNet50 pipeline was followed by the traditional ML-based data processing (PCA).

The discrimination between meningiomae on the basis of molecular markers was also assessed by several ML-based methods without proper description of parameters applied and the selection of the method performed.

The results were assessed via limited number of performance indicators. Several critical indicators were missed, including F1 scores.

The mathematical descriptions of the applied algorithms or references for such descriptions are missed.

To ensure the reproducibility of the developed solution, the authors must improve the manuscript extensively.

The critical issue

-----

[1]

The authors declared that “the requirement for informed consent for this study was waived because of its retrospective nature.”

Despite the claim, informed consent is one of the founding principles of research ethics. Consent should be obtained before the participant enters the research prospectively.

Researchers should ensure that they comply with the General Data Protection Regulation (GDPR) during and after the consent process. Where research includes “filming” or “photography”, specific guidance in the Photography and GDPR toolkit should be referred.

Please read https://researchsupport.admin.ox.ac.uk/governance/ethics/resources/consent

and other relevant sources to understand legal issues relevant to informed consents.

Due to legal issues, the following changes are critical for this manuscript.

[1a]

Due to the claims, it is indispensable to include the text of the document approving the study protocol to the supplement of this manuscript.

[1b]

According to general ethics guidelines (the Declaration of Helsinki and Personal Data regulations like GPDR), it is highly recommended to gain the written informed consent from every alive participant enrolled to this study.

[1c]

In case of mentally disabled/deceased participants, it is highly recommended to obtain the “written informed consent” from their Legal representatives / heirs on the prospective use of personal data in this study.

[1d]

LL103-104 - “protocol was approved by the XXXX University … Ethics Committee” – Please identify the name of the University.

Major issues

-----

[2]

In the relevant subsection of the manuscript, please clearly indicate the reasons of not including into analysis any MRI images generated by protocols different from SWI, to form the multimodal dataset before the analysis.

-----

[3]

Please describe all modifications applied to the ResNet50 entity in full.

[3a]

In particular, please describe aims achieved by ResNet50 modifications applied within the relevant subsection of the manuscript (2.7. Deep-learning-based feature extraction).

[3b]

In the text of the relevant subsection, please also reflect whether other layers of the ResNet50 pipeline were altered and the consequences described.

[3c]

Please depict the ResNet50 pipeline graphically (in a form of the Figure), indicating the changes applied by the authors to the ResNet50 architecture and including the Figure into the manuscript text within a relevant subsection.

To ensure clear representation of changes applied to the ResNet50 pipeline, please consult with Elpeltagy, Sallam (2021) or any other similar paper.

Elpeltagy, M., Sallam, H. Automatic prediction of COVID− 19 from chest images using modified ResNet50. Multimed Tools Appl 80, 26451–26463 (2021). https://doi.org/10.1007/s11042-021-10783-6

-----

[4]

The Figure 3 does not provide the clear linear representation of the developed pipeline and was not refereed properly in the relevant subsections of the manuscript. The representation of the whole workflow must be improved greatly to ensure clear representation of the analysis workflow, identifying clear correspondence between the text details and the Figure 3 elements.

[4a]

Please improve the size of the texts and the quality of the scheme on the Fig. 3.

[4b]

It is also possible to use the Y-axis as a global time axis discriminating the timeline of pipeline steps and substeps performed.

[4c]

Every tested method must be indicated at the relevant stage of the pipeline, where it had been tested.

-----

[5]

The relevant math is not described within the text of the manuscript.

Within the text of the manuscript, please describe all critical math steps applied to data or provide relevant citations on the math underlying the developed solution.

[5a]

In the supplement (as a text file), please provide the algorithm parameters applied in pair with every multiple classification algorithm enrolled in this study

[5b]

Within the text of the relevant subsection, please describe in details the preprocessing procedure involving the isolation forest technique, with criteria identified to exclude 5 NF-2-related instances and 10 S100-related instances.

[5c]

The synthetic minority oversampling technique (SMOTE) must be reported in full details within the text.

[5d]

Since the PCA was applied to reduce the dimensionality of the features, the authors must report the scree plot gained in the supplement.

-----

[6]

Demographic and clinical characteristics of the dataset must be visualized as Raincloud plots with jittering, the Median, IQR boxes and CI95 intervals visualized for each quantitative characteristic like age, between control and affected samples.

[6a]

On the figure panel jitter plots, the excluded cases must be indicated as individual points distinct from the general dataset.

-----

[7]

The results of the dataset changes achieved via the SMOTE procedure must be indicated directly within the text of the manuscript and depicted in the supplement as an additional Figure.

-----

[8]

The authors claimed that “Chi2 test and Fisher’s exact test could not identify any significant associations between NF-2 mutations or S100 protein expression and various morphological features”

Moreover, the authors did not report the estimations of variance for the ROC curves reported on the panels of the Fig.4.

Several improvements are required to ensure clarity on the discriminations tasks reported by the authors:

[8a]

Within the text, please identify whether the 2x2 variant or some other variants of the Chi2 and Fisher exact tests were applied.

[8b]

The authors must estimate the variance of the ROC curves reported on the panels of the Fig.4, using cross-validation or another approach.

[8c]

It is highly recommended to analyse associations between demographic/radiological characteristics and molecular markers using exploratory PCA technique, reporting the scree plot (as a Figure) and other relevant findings reflecting large associations between characteristics and molecular markers analysed.

If some of the analysed characteristics form group associations with identified principal components (PCs), that PCs also should be reported (as Figures)

[9]

The data presented in a form of the Tables 2-5 must be visualized as Figures in the main text of the manuscript (Raincloud plots with jittering, the Median, IQR boxes and CI95 intervals reported graphically), moving the Tables 2-5 to the supplement.

-----

[10]

The results of testing various ML algorithms must be reported in the Supplement in a form of the table, comparing every achieved performance indicator and the variance of such indicator estimated by cross-validation.

[10a]

The indicators must be reported as Mean + SD or Median + IQR.

[10b]

Within the text of the manuscript, the authors must identify the underlying basic algorithms applied for every ensemble algorithm enrolled (Adaboost, LGBM etc).

[10c]

For every algorithm, the authors must also report the following performance indicators (Mean + SD or Median + IQR), comparing tested methods on the basis of those indicators:

- precision

- F1 score

[11]

In the discussion section, please provide comparisons with SOTA models (if possible).

If not possible, please provide the explanation within the text of the manuscript.

Minor issues

[12]

L137 In the text, please identify dose (or dose ranges) of Gd-DTPA applied to patients.

[13]

Please include the scale bars on the Fig 2 panels.

Comments on the Quality of English Language

Language issues

[14]

A number of clauses and sentences are clear or even fuzzy and must be improved.

Several examples are below:

LL52-53 - “biological behavior of meningiomas” - Please clarify

L54 “have a lower grade” (of what?) - Please clarify

L54 “longer progression-free survival” (of people? meningiomas? smth else?) - Please clarify

LL56-57 “However, the identification process (of ?) necessitates conducting tissue diagnosis.” – Please clarify the sentence.

L91 “radiomics features derived from deep learning architectures” – Please clarify

L94 “which reflects” – reflecting? Please consider to improve

LL95-96 “features were more effective” – Please clarify

LL191-192 (and other cases of occurrence) “univariable and multivariable logistic regression” univariate? multivariate? – Please clarify.

[15]

Within a single sentence, please avoid using the same term in a raw.

Some examples

L53 “meningiomas … meningiomas” –

L63 “MRI … MRI”

Reviewer 2 Report

Comments and Suggestions for Authors

Study proposed using radiomics and morphological features of susceptibility-weighted MRI (SWI) to investigate the predictive power of machine learning models for neurofibromatosis type 2 (NF-2) mutation and S100 protein expression in Meningiomas. Different features such as tumor growth pattern, peritumoral edema, sinus invasion, hyperostosis, bone destruction, and intratumoral calcification were semi-quantitatively assessed and compared to the deep learning features. Although the study is interesting, there are fundamental aspects that authors need to address:

  1. The writing needs to be simplified as it hard to understand what the author did even by just reading the abstract. The goal is unclear with a lot of results hence hard to read. Is this a multi-class classification? it simply a guess but not really mentioned.
  2. A dataset summary table should be included. Was the registration done manually or can FSL perform that automatically?
  3. In section 2.7, there is lack of details on how the transfer learning in performed. On which data? splits proportion? learning rate etc… Which pretrained weights was used?
  4. It wrong to standard the data before splitting into training/ testing. Standardization across instances should be done after splitting the data between training and test set, using only the data from the training set. This means you standardize the test set using the parameters previously obtained when standardization the training set.
  5. In line 223, authors mentioned that the extracted features were preprocessed by standardization, removing outlier etc.., Are these features from the deep learning ? if so how do they differentiate between NF-2 mutation features and S100 protein expression features?
  6. Based on the study description, there is a high chance of data leakage between training and testing. Hence authors should provide a section detailing the data description and the splitting before moving in to modelling. How many images used for training, how many feature extracted, what about before and after SMOTE and PCA? Because multiple problem are addressed but the details are missing.
  7. It is unclear or confusing on what the authors are trying to achieve. It is looking at CNN vs other hand-engineered features? Are these features coming from the Pyradiomics? Since the package also offer deep learning features so authors need to justify why not using them etc.
  8. The proposed pipeline in figure 3 is far from complete as it still confuse the reader of the purpose of the study. I understand that the purpose may not be to compare many algorithms but that how the work is presented at the moment.
  9. Authors should also be aware of optimizing each method separately and avoid blindly running these algorithms with default parameters. Existing optimization methods can be useful to address this challenge such as grid search or more optimized approach with framework such as optuna.

Round 2

Reviewer 1 Report

Comments and Suggestions for Authors

The manuscript has been improved extensively by the authors.

Only several minor issues are remained to be resolved.

Minor issues

[1]

In the Abstract section, please provide the readers with the uncertainty estimates for the AUCs provided.

[1a]

p-values might be removed from the Abstract section as an excessive information.

[2]

Please consider to place Tables S6 and S7 as Tables in the main manuscript, moving Figs. 6-11 to the Supplement as supplement figures.

[2a]

In the Tables S6 and S7, please place the AUCs as the first performance indicator.

Please also sort the rows in the Tables S6 and S7 according to the AUC values, in the descending order.

[2b]

When the range for the performance indicator is less than 0.01, please indicate the specific value, not a result of its fixed rounding.

For example, “accuracy = 0.49 ± 0.00” [L506]

In this case, please indicate the real value for 0.00

Like “accuracy = 0.49 ± 0.002”

[2c]

Please also check Tables S6 and S7 for such rounded values equal to 0.00 and improve when needed.

[2d]

In the tables S6 and S7, are the AUC values of 0.00±0.00 correct for the “SVM - Linear Kernel” and “Ridge classifier”?

Please check and discuss in the Discussion section, if needed.

[3]

In the captures for the Figs 4-11, please indicate the parameters displayed on the panels (Means or Medians, SDs or IQRs, etc)

[4]

On the panels of the Figs 8-11, the text must not overlap with the graphical elements.

Please consider to position the p-values and odds ratios under the relevant graphical panels.

[5]

In the capture for the Fig. 15, please indicate the type of uncertainty band used.

Comments on the Quality of English Language

Language issues

[6]

Please check the text for unresolved fuzzy clauses.

Some examples are below

[L255] “a combination of bootstrapping and cross-validation (?)” procedures? extensions? Please clarify.

[L278] “tumor area within (?) rectangular regions” which rectangular regions? Please clarify.

[7]

In the capture for the Fig.12, please use “The” for the panels b and d

“(The) test set result”, since these results are unique.

Reviewer 2 Report

Comments and Suggestions for Authors

Thanks to the authors for updating the manuscript based on the provided feedback. While the manuscript has been substantially improved, few points need to be further clarified below:

- "For the NF-2 copy number loss dataset, after outlier removal and SMOTE application, the training set initially contained 69 cases, which increased to 72 cases." I am not sure of the added value of using SMOTE technique in this settings, authors need to justify this (69 to 72 features)?

- As the title suggested there is a need to include an end-to-end information of the study in the Fig 3. For example, I would breakdown the pipeline as follows: 1. inputs, 2. feature extraction, 3. Feature preprocessing/optimization, 4. classification etc.  It should be a graphical abstract of the study.

- To my understanding the study aimed to investigate the added value of the proposed pipeline in predicting meningiomas patients with or without NF2 copy number loss and also with or without S100 expression. So it is a binary classification pipeline but proposed for the two cases (i.e. NF2 copy number loss and S100 expression) ? This need to be reflected in figure 3.  
